# Foliar Spraying of Selenium Combined with Biochar Alleviates Cadmium Toxicity in Peanuts and Enriches Selenium in Peanut Grains

**DOI:** 10.3390/ijerph19063542

**Published:** 2022-03-16

**Authors:** Shiwei Shao, Bing Ma, Liuhuan Ai, Xia Tian, Lei Zhang

**Affiliations:** School of Environment and Science, Qingdao Agriculture University, Qingdao 266109, China; shaosw@stu.qau.edu.cn (S.S.); mbing____lw@163.com (B.M.); 18211756563@163.com (L.A.); 200601139@qau.edu.cn (X.T.)

**Keywords:** cadmium, selenium, glutathione, phytochelatin, subcellular distribution

## Abstract

Cadmium (Cd) pollution in soil, particularly in peanut production, is a problem that has attracted global concern and needs solutions urgently. Selenium (Se) can alleviate Cd toxicity; however, the underlying mechanisms are not completely understood. Therefore, two varieties of peanut (*Arachis hypogaea* Linn.), “Huayu 23” and “Huayu 20”, were chosen as the target crops for this study. A pot experiment was conducted to investigate the effects of two Se application methods combined with biochar on the accumulation of Cd and Se, and the best application method was identified. In addition, the role of Se in alleviating Cd toxicity in peanuts was studied. The results indicated that both Se and biochar decreased the Cd content in peanuts and alleviated Cd toxicity. However, the combined application of foliar Se and biochar significantly increased the peanut biomass by 73.44–132.41%, increased the grain yield of Huayu 23 by 0.60–1.09 fold, and Huayu 20 by 2.38–3.48 fold. Additionally, Cd content in peanut grains was decreased by 32.81–50.07%, and Se content was increased by 31.57–99.75 folds. Biochar can decrease the absorption of Cd from the soil, while Se can increase the accumulation of Cd in cell vacuoles by increasing glutathione and phytochelatin to decrease the movement of Cd into the grains. Therefore, our results indicate that the combined application of foliar Se and biochar can effectively promote the enrichment of Se in peanuts and suppress Cd toxicity.

## 1. Introduction

Cadmium (Cd) is one of the most toxic pollutants in soil. Due to its high mobility, toxicity, and long-term and irreversible characteristics, it causes great harm to farmland soil [1]. When compared with other metals, Cd is easily absorbed by plant roots and transferred to the above-ground parts, affecting the normal growth of plants by accumulating in the reproductive organs and edible parts [2]. Studies have indicated that Cd affects the absorption of beneficial trace elements by plant roots [3]. In plants, Cd can damage photochemical organs, change the structure of chloroplasts, and hinder photosynthesis [4,5,6]. Cadmium induces plants to produce reactive oxygen species, which indirectly or directly destroy plant cell molecular functions. Excessive reactive oxygen species can cause cell membrane rupture and cell death [7]. In addition, Cd can damage cell organelles and ultrastructures. Moreover, heavy accumulation of Cd in the edible parts threatens human health [8]. It has been confirmed that daily exposure to 30–50 mg Cd greatly increases the risk of diseases, such as cancer, renal dysfunction, and hypertension [9,10]. Excessive accumulation of Cd in oil and grain crops and entry into the human body through diet is the main mechanism by which Cd causes harm to humans.

Peanut (*Arachis hypogaea* Linn.) is the second largest edible oil crop in China, and its planting area is second to rape [11]. Studies have indicated that the enrichment coefficient of Cd in peanut kernels is 4.5 times that of rape [12]. In contrast, Cd ranks third after zinc and copper in terms of the degree of heavy metal enrichment in peanuts [13]. According to the World Health Organization and World Allergy Organization, the maximum permitted concentration for Cd in peanuts is ≤0.1 mg·kg^−1^, and consumption of peanuts with a Cd concentration of more than 0.2 mg·kg^−1^ can lead to certain risks in public health. Therefore, it is particularly important to identify an effective method to decrease Cd content in peanuts.

Selenium (Se) is an essential trace element in humans and animals, and plays a key role in human health. It is responsible for enhancing human immunity, and preventing various diseases and aging. However, several studies report inadequate Se intake in large populations worldwide [14,15,16]. Although the role of Se in plants remains elusive, research has revealed that Se at an appropriate concentration can promote plant growth, improve resistance to oxidative stress, and increase photosynthesis [17,18]. Moreover, Se can effectively mitigate pests, diseases, plant senescence, and heavy metals, particularly in Cd-stressed plants [19]. This antagonistic effect is reflected in several crops, such as wheat, rice, and rape [20,21,22]. Studies have indicated that proteins in plants can be combined with inorganic Se and converted into nontoxic organic Se for absorption by the human body [23]. Moreover, the protein content of peanut kernels accounts for 25–36% of its dry weight, and is the main source of protein for human consumption [24,25]. Thus, Se may be effective in inhibiting Cd toxicity in peanuts and can supplement the Se required by the human body through peanuts.

Selenium application can be performed in four methods as follows: seed dressing, seed soaking, soil application, and foliar application [26,27,28,29]. The use of sodium selenite or sodium selenate can significantly increase Se content in crops. Studies have indicated that applying appropriate amounts of Se to Cd-treated soil can significantly decrease the amount of exchangeable Cd in the soil. Feng et al. [30] suggest that this is due to the formation of Se–Cd heavy metal complexes between Se and heavy metals in the soil, which decreases the effectiveness of Cd. In plants, Se can reduce the transportation of Cd and decrease the movement of Cd to reproductive organs, such as grains [31]. In addition, studies have found that Se can increase the levels of antioxidants, such as enzymatic superoxide dismutase, catalase, peroxidase, glutathione reductase, and glutathione (GSH), thereby inhibiting the transport of Cd in plants [32]. However, in the case of severe Cd pollution, Se alone is not effective in mitigating Cd. Therefore, other materials in combination with Se fertilizer can be a good alternative to remediate Cd pollution.

Biochar has good adsorption and stability characteristics, and plays an important role in improving the physical and chemical properties and soil fertility. Moreover, biochar can repair heavy metal pollution in the soil [33,34,35]. It can decrease the bioavailability of heavy metals in the soil through adsorption, complexation, precipitation, etc., and plays a significant role in alleviating stress due to heavy metals [36]. However, the combined treatment method of biochar and Se fertilizer to decrease Cd content in peanuts, and its potential interaction and mechanism of action, remains elusive.

Therefore, in this study, we used peanuts as the target crop to investigate the following issues using greenhouse pot experiments: (1) effect of biochar + Se fertilizer on peanut Se enrichment and Cd suppression; (2) the effects of foliar Se + biochar and soil Se + biochar in enriching Se and inhibiting Cd, and identification of the best combination of biochar and Se fertilizer; (3) the subcellular distribution and microdistribution of Cd to expound the mechanism by which Se regulates Cd accumulation in peanuts. 

## 2. Materials and Methods

### 2.1. Experimental Materials

The test soil was collected from a 0–20 cm plough layer of farmland soil in Chengyang District, Qingdao. Stones and plant residues were removed using a 5 mm sieve and air-dried. The original soil had a pH of 8.66 and contained 0.85 mg·kg^−1^ Cd, 0.0025 mg·kg^−1^ Se, 67.50 mg·kg^−1^ alkaline hydrolysis nitrogen, 4.19 mg·kg^−1^ available phosphorus, 161.90 mg·kg^−1^ available potassium, and 15.90 mg·kg^−1^ organic matter. We selected two varieties of peanut, “Huayu 20” (high Cd accumulation) and “Huayu 23” (low Cd accumulation) [37]. Corn stover biochar was produced in a muffle furnace at 550 °C. The experimental equipment was soaked in 2.00 mol·L^−1^ HNO_3_ solution for 48 h before use, washed thrice with tap water and deionized water, and dried for later use. Seeds with full grains and uniform size were selected, washed, dried, soaked in 30% H_2_O_2_ solution at 25 °C for 30 min, and germinated in an artificial climate box at 28 °C. After germination, we selected seedlings with good vigor and uniform size. Five plants per pot were transplanted into polyethylene plastic pots. After the crop emerged, thinning was performed, and three plants of similar growth were retained in each pot.

### 2.2. Potted Peanut Experiment

The pot experiment was conducted in the scientific research intelligent greenhouse of Qingdao Agricultural University. The greenhouse allowed the entry of natural light and maintained a humidity of 45%, temperature of 27 °C during the day, and 21 °C at night. The polyethylene plastic pots were 18 cm high with an inner diameter of 28 cm. Each pot contained 6.3 kg of soil. Dosages of 0.15 g N/kg soil, 0.15 g P_2_O_5_/kg soil and 0.15 g K_2_O/kg soil were utilized as a base fertilizer. The compound fertilizer N—P_2_O_5_—K_2_O: 15—15—15 and biochar was applied along with the base fertilizer at a dosage of 1% (*w*/*w*). Cd was supplied as CdCl_2_·2.5H_2_O with two levels: 5 and 20 mg·kg^−1^ of soils. Low concentration of cadmium (5 mg·kg^−1^) was 8 times the third-level standard for soil cadmium pollution, to simulate a low-cadmium-pollution environment; the high concentration of cadmium (20 mg·kg^−1^) was to simulate soil conditions under high cadmium stress. Six treatments were set up: blank (CK), biochar (T1), soil Se (T2), soil Se + biochar (T3), foliar Se (T4) and foliar Se + biochar (T5). Each treatment was repeated three times per group.

The soil selenium was applied with sodium selenite (Na_2_SeO_3_) solution at the dosage of 0.5 mg·kg^−1^ along with the base fertilizer. The amount of Se sprayed on the foliar surface was the same as that sprayed on soil, that is, thrice in the seedling stage, once every three days, and sprayed with other groups of peanut plants with the same amount of deionized water. The peanut in the pot was cultivated in the greenhouse for five months and air-dried for later use. After collecting fresh samples of peanut plants, the remaining samples were dried at 105 °C for 15 min, further dried at 80 °C to achieve constant weight, and ground for further analysis.

### 2.3. Determination of Total Cd and Se Content in Plants

To determine the concentration of Cd in plant tissues, the plant samples were digested in 10 mL mixed acid (9:1 ratio of nitric acid:perchloric acid, *w*/*w*) during 12 h, followed by digestion on a hot plate. The samples were heated until a clear solution was obtained, and heating was continued until 2 mL of the solution was remaining. The solution was then diluted to 50 mL with deionized water. The supernatant was subjected to inductively coupled plasma emission spectrometry (OPTMA8000DV, PE, Waltham, MA, USA) to measure Cd concentration.

To determine Se concentration in plant tissues, the samples were digested in 10 mL mixed acid (9:1 ratio of nitric acid:perchloric acid, *w*/*w*) during 12 h, followed by digestion on a hot plate. The samples were heated until they became clear and colorless and were emitting white smoke, and heating was continued until 2 mL of the solution was remaining. The solution was then cooled, and 5 mL hydrochloric acid was added. Heating was continued until the digestion solution was clear and colorless and emitting white smoke. After cooling, the solution was diluted to 50 mL with deionized water. The supernatant was subjected to hydride atomic fluorescence spectrometry (AFS-933, Beijing, China) to determine Se concentration.

The blank and a standard reference material (GSB-27, Chinese onion) were both included in the digestion process to verify the accuracy and precision of the sample analysis, and the recovery for GSB-27 was between 91% and 105%

### 2.4. Extraction of Subcellular Components from Peanut Leaves Using Differential Centrifugation

A leaf sample (approximately 0.20 g) was ground into a homogenate with a precooled extract. The composition of the extract was as follows: 0.25 mol·L^−1^ sucrose, 0.1 mol·L^−1^ Tris-HCl (pH 7.55), and 1 mmol·L^−1^ dithiothreitol (C_4_H_10_S_2_). The homogenate was centrifuged at 150× *g* for 10 min, and a cell-wall-containing residue (component F1) was obtained. The supernatant was centrifuged at 3000× *g* for 15 min, and a cell nucleus and chloroplast fraction (component F2) was obtained. The supernatant was further centrifuged at 10,000× *g* for 20 min, and the precipitate containing mitochondria (component F3) and supernatant containing the cell soluble fractions, namely, vacuoles, ribose, and nucleoprotein (component F4), were obtained. The specific steps of digestion and determination of the extracted components were the same as those for the determination of Cd and Se content.

### 2.5. Scanning Electron Microscopy (SEM) and X-ray Energy Spectrum (EDS) Analysis

A fresh sample was fixed in 2.5% glutaraldehyde solution at 25 °C for 1–2 h, transferred to 4 °C, and fixed overnight. The fixative solution was removed, and the samples were rinsed thrice with freshly prepared 0.1 M phosphate-buffered saline (PBS pH 7.4) for 15 min. After rinsing, the sample was fixed with 1% osmic acid solution for 1.5 h, and the waste osmic acid solution was carefully discarded. The sample was rinsed thrice with 0.1 M PBS (pH 7.4) for 15 min and dehydrated sequentially in gradient concentrations of ethanol (30%, 50%, 70%, 80%, 90% and 95%) for 15 min each. The 95% ethanol solution was removed, and the sample was treated with 100% ethanol twice for 20 min. The sample was dried at the critical point, coated, and visualized using SEM (Nova Nano 450, FEI, now under Thermo Scientific (Waltham, MA, USA) observation).

### 2.6. Statistical Analysis

The following formula was used to calculate the accumulation of Cd in roots, stems, leaves, and grains, and the transfer coefficient from root to stem, stem to leaf, and leaf to grain:(1)$$\mathrm{ACd}\;=\;\mathrm{CA}-\mathrm{Cd}\;\times {\;A}_{\mathrm{biomass}}$$
(2)$$\mathrm{TF}=\frac{\mathrm{CA}-\mathrm{Cd}}{\mathrm{CB}-\mathrm{Cd}}\times 100\%$$
where ACd represents the accumulation (μg) of Cd, CA − Cd and CB − Cd represent the Cd content (mg·kg^−1^), and A_biomass_ represents the biomass (g) of Cd in roots, stems, leaves, and grains of peanuts.

Statistical analyses were performed using one-way analysis of variance and the least significant difference (LSD) method using the SPSS 20.0 software, IBM, Armonk, New York, USA. The variability of the data was expressed as the standard error (*p* ≤ 0.05). The correlation analysis was based on Pearson’s correlation coefficient, and *p* ≤ 0.01 and *p* ≤ 0.05 were considered statistically significant. All tables and graphs were created using Microsoft Excel software.

## 3. Results

### 3.1. Effect of Different Treatments on Peanut Biomass

The biomass of peanut roots, shoots, and grains under different treatments is shown in Table 1. When compared with the control, T1, T2, T3, T4 and T5 treatments significantly increased the root and above-ground biomass and grain yield of peanuts. In Huayu 23, the highest grain yield was observed in the T3 group, and compared with the control, the grain yield increased by 109.30% and 59.81% in the high and low Cd concentrations, respectively. The yield in other treatment groups increased considerably; however, the degree of improvement was different under different Cd concentrations. In general, the effect of Se on the growth of Huayu 23 was significantly higher than that of biochar, and the combined application of Se and biochar had a superior effect on the growth of peanuts affected by Cd pollution.

The highest grain yield of Huayu 20 was observed in the T5 group, which was increased by 347.50% (high Cd concentration) and 237.50% (low Cd concentration) when compared with that in the control group. Selenium was more effective than biochar in Huayu 20, and the combined effect of the two was significantly higher than that of a single application. Similar results were observed in Huayu 23. Meanwhile, T5 treatment was more effective in increasing Huayu 20 grain yield, and its effect was higher in Huayu 20 than in Huayu 23. This may be associated with the different Cd absorption capacities of the peanut varieties. Based on the analysis of the root and above-ground biomass and grain yield of the two peanut varieties Huayu 23 and Huayu 20, it was concluded that the combined application of Se and biochar had an alleviating effect on peanut growth under Cd stress, and the effect was superior to single treatment. Comparing the effect of two selenium application methods on improving peanut yield, the combined application of foliar selenium application and biochar was better.

According to variance analysis, all three showed substantial effects on peanut grain yield, and the two-factor interaction and three-factor interaction among the three had significant effects on peanut grain yield. 

### 3.2. The Content of Cd and Se in Peanuts

Table 2 lists the Cd content in roots, stems, leaves, and grains of the two peanut varieties at high and low Cd concentrations. At high (20 mg·kg^−1^) and low (5 mg·kg^−1^) Cd levels, the order of Cd content in roots, stems, leaves, and grains of Huayu 23 and Huayu 20 treatment groups was as follows: leaf > root > stem > grain. This sequence was expressed in all treatments, except for the high Cd soil Se treatment in Huayu 23. Under the same Cd levels of the two peanut varieties, the Cd content in each organ of Huayu 20 without treatment was higher than that of Huayu 23, which was related to the sensitivity of peanut varieties to Cd. According to the results of a variance analysis, peanut variety also played a major role in reducing cadmium stress. 

In a single cultivar polluted with the same concentration of Cd, the Cd content in roots, stems, leaves, and grains of each treatment group was lower than that of the blank group. They exhibited similar behaviors in different cultivars and Cd concentrations. Under high Cd levels, the Cd content in the roots, stems, leaves and grains of Huayu 23 decreased by 22.37%, 50.46%, 27.49% and 47.87%, respectively, which was the lowest under T5 treatment. Under high Cd pollution level, the Cd content of each organ in Huayu 20 was similar to that in Huayu 23. The lowest Cd content was observed under T5 treatment, and the Cd content in roots, stems, leaves, and grains was 62.11%, 54.81%, 46.67% and 50.07% lower than that of the blank, respectively. Under a single T1 treatment, the Cd content in the grains of Huayu 23 and Huayu 20 was significantly decreased, indicating that biochar played a role under high Cd concentrations. However, the two different Se application methods and the combined application of Se and biochar produced different results on the reduction of Cd content in the grains of Huayu 23 and Huayu 20. This may be related to the different effects of Se absorption in different peanut varieties.

In the low-Cd-concentration environment, the cadmium content in roots, stems, leaves, and grains of T5 treatment group was significantly lower than that of other treatment groups. This performance was the same as the high-cadmium-pollution treatment, and the decreasing trend of cadmium content between each treatment group was similar. According to the results of a variance analysis, biochar played an effective role in alleviating Cd stress in peanuts, and this effect was further enhanced after the addition of Se. However, application of Se using different methods showed varying effects for different peanut varieties, but the performance trends under different treatments were consistent. The cadmium content in peanut grains was significantly affected by the addition of biochar and selenium application and peanut varieties. Although the interaction effect between variety and biochar was not significant, the interaction between variety and biochar, biochar and selenium, and the interaction among the three were all significant. Under the low cadmium level, the two-factor interaction among peanut varieties, biochar, and selenium had a significant effect on the cadmium content of peanut grains, but the three-factor interaction was not significant. 

Selenium content in the roots, stems, leaves, and grains of peanuts under different treatments is shown in Table 3. The four groups T2–T5 were treated with Se, and the content of Se in the roots and stems was significantly higher than that of the control and T1. This indicates that both soil and foliar Se can increase the accumulation of Se in peanuts. Among the leaves and grains of peanuts, the Se content in plants treated with foliar Se (T4 and T5) was significantly higher than that of other treatments. In contrast, the Se content in peanut leaves treated with soil Se (T2 and T3) was slightly higher than that in the control group, with no significant difference. In addition, the Se content in Huayu 20 grains under high Cd concentration did not exhibit a significant difference when compared with the control. Moreover, Se content in other grains was significantly higher than that of the control; however, it was significantly lower than that of grains of peanut subjected to foliar Se spraying. Upon comparison of the foliar Se treatment before and after adding biochar, the Se content in the grains of the two peanut varieties treated with foliar Se before adding biochar was significantly higher than that of the latter. This indicates that the addition of biochar affects the Se content in peanut kernels. Since the Se content in the grains treated with soil Se was low, the Se content did not change significantly after the addition of biochar. However, it can be concluded that the addition of biochar reduced the Se content in the grains of Huayu 23 and Huayu 20 under T2 and T3 treatments. The statistical results showed that under low cadmium pollution, peanut variety, biochar, and selenium, and the interaction of two or three factors, had significant effects on grains selenium content. However, under high-cadmium-pollution conditions, the interaction between varieties and biochar and the three factors was not significant.

### 3.3. Accumulation and Transportation of Cd in Peanuts

Addition of biochar and Se significantly affected the transport (Table 4) and accumulation (Figure 1) of Cd in peanuts. As shown in Figure 1, Cd was mainly concentrated in the stems and leaves, followed by the roots and grains. As shown in Figure 1a, addition of Se increased the relative proportion of Cd accumulation in roots and decreased the transport of Cd to the stem. This was consistent with the change in TF_Root–Shoot_ of Huayu 23 under high Cd concentrations (Table 4). The addition of Se significantly decreased TF_Root–Shoot_, and foliar Se treatment was significantly more effective than soil Se treatment. Moreover, the addition of biochar and selenium increased the relative proportion of Cd accumulation in roots, and Se treatment significantly decreased the transport of Cd from the leaves to the grains, thereby decreasing Cd content in the grains (Table 1). At low Cd concentrations, the accumulation of Cd in the roots did not change significantly. Similarly, there was no significant difference in Cd accumulation in Huayu 23 roots, except under T3 treatment. In contrast to high Cd concentration, TF_Leaf–Grain_ under T2 treatment in low Cd concentration was significantly lower than that in other treatment groups, reducing the accumulation of Cd in grains.

Cadmium transport in Huayu 20 under each treatment was different from that of Huayu 23. There was no obvious trend in the transport of cadmium among different treatments, but the accumulation of selenium and biochar could increase the accumulation of cadmium in roots (Figure 1b,d).

### 3.4. Subcellular Distribution, GSH Content, and PC Content of Cd in Peanut Leaves

In all treatments, the distribution ratio of Cd in different subcellular components of peanut leaves was as follows: cell wall (F1) > soluble cytoplasm (F4) > mitochondria (F3) > cell nucleus and chloroplast (F2) (Figure 2). In the untreated CK group, more than half of the Cd was absorbed by the cell wall, and the addition of Se significantly affected the subcellular distribution of Cd in the peanut leaves. Selenium decreased the absorption of Cd by the cell wall and increased the absorption of Cd by the soluble cytoplasm; however, the extent of this effect depended on the use of Se. In contrast, spraying Se fertilizer alone significantly increased the percentage of Cd in the soluble cytoplasm. This result is reflected in two pollution concentrations of the two peanut varieties. The high and low Cd concentrations in Huayu 23 increased by 37.66% and 24.26%, respectively, while the high and low Cd concentrations in Huayu 20 increased by 19.46% and 45.85%, respectively. Although the two varieties of peanuts showed similar results under the same treatment, the magnitude of increase under different concentrations of Cd pollution differed.

Table 5 and Table 6 show the GSH and phytochelatin (PC) content of peanut leaves, respectively. Under high Cd concentrations, spraying Se fertilizer on foliage significantly increased the GSH content in Huayu 23 and Huayu 20 leaves. However, the difference in GSH content among the other treatments was not significant. As observed under high Cd concentration, foliar Se significantly increased the GSH content in leaves under low Cd concentrations. The difference was that soil Se significantly increased the GSH content, compared to content in the CK group. The addition of biochar alone did not significantly affect the GSH content; however, a combination with Se treatment decreased the GSH content. The change in the content of PC was the same as that of GSH; however, the addition of biochar further decreased the content of PC in Huayu 23. This effect was not observed in Huayu 20.

### 3.5. SEM–EDS Analysis of Peanut Leaves

We selected three treatments, CK (Figure 3a), T4 (Figure 3d), and T2 (Figure 3g) under high Cd concentration in Huayu 23 for SEM and EDS analysis (Figure 3c,f,i). As shown in the figure, the longitudinal section of peanut leaves was observed using SEM. Under different treatments, the surface of the mesophyll cells did not appear damaged or wrinkled, indicating that the Cd stress did not affect the normal growth of peanut under the Cd-contaminated soil environment.

The red dots in the figure represent the selection points for energy spectrum analysis, and the leaf mesophyll cells were selected for the three treatments. In the mesophyll cells, the relative mass of carbon accounts for approximately 50%, followed by oxygen and nitrogen. In the CK group (Figure 3b), the relative masses and atomic weights of Cd were higher than those of Se, and the ratio of the relative amount of substance of Cd to Se was 1.29. Following the addition of Se, the relative amount of Se increased, while that of Cd decreased. The relative amount of substance ratios of Cd and Se in T4 (Figure 3e) and T2 treatments (Figure 3h) decreased to 0.42 and 0.67, respectively. When compared with CK, foliar Se increased the relative quality of the trace elements magnesium (Mg) and silicon (Si) in the mesophyll cells, while soil Se increased the relative quality of Mg, Si, sulphur (S) and iron (Fe).

## 4. Discussion

In previous studies, peanut plant biomass and grain yield were used as important indicators for evaluating peanut tolerance to heavy metal stress [38]. The addition of Se fertilizer and biochar significantly increased peanut biomass and grain yield, which is consistent with the results of previous studies [39]. Moreover, we found that the combined application of Se and biochar can further increase the grain yield of the two peanut varieties Huayu 23 and Huayu 20, based on a single treatment. The combination of the two significantly decreased the Cd content in the peanut tissues, indicating that Se and biochar played an effective role in alleviating Cd stress. Previous studies have indicated that biochar can repair heavy metal pollution in soil and effectively decrease the bioavailability of soil Cd, thereby reducing the absorption of Cd in the soil by plants [40].

Selenium’s inhibition of cadmium absorption has been confirmed in many plants [41,42,43,44]. In this study, the Cd content in the grains of Huayu 23 under the two Cd concentrations decreased by 24.59% and 24.78%, respectively, with the addition of biochar alone. This indicated that biochar had the same effect on reducing the Cd content in grains. Under high and low Cd pollution, foliar Se + biochar treatment resulted in the lowest Cd content in grains, indicating that the mitigation effect of Se on Cd may be related to the Cd content. Therefore, we conducted a correlation analysis on the Se and Cd content in each organ of the peanut plant. Table 7 indicates that the Se content and Cd content were negatively correlated. This result was consistent with that of Guo et al. [31] in their study on rice. This indicates that in peanut plants, an increase in Se content can decrease Cd content in the corresponding organs and thereby decrease the absorption of Cd by peanuts.

The combined use of biochar and foliar Se was the best method to decrease Cd content in Huayu 20. In addition, the combined application of soil selenium and biochar was second only to foliar selenium + biochar in reducing peanut cadmium content. Huang et al. [45] revealed that the application of exogenous Se to soil can decrease the bioavailability of Cd in the soil. This was associated with the formation of Se–Cd complexes between the selenite product in the soil and Cd, and Se affected the rhizosphere microenvironment [46,47]. This was reflected in both Huayu 23 and Huayu 20. Following the addition of Se to the peanut, the foliar Se + biochar effectively decreased the Cd content in the above-ground biomass and grains of peanuts, and its effect was significantly higher than that of soil Se + biochar.

The application of Se not only decreased the absorption of Cd by peanuts but also decreased the transport of Cd to the grains. In previous studies, the addition of Se effectively decreased the transfer of Cd from the roots to the shoots [48]. However, in our study, this was only found in Huayu 23, which was polluted with high Cd concentrations. In the other three groups, the Cd transport coefficient did not show similar results. Huayu 23 is a cultivar with low Cd accumulation and moderate Cd tolerance. The Cd transport coefficient was consistent with the Cd accumulation, and addition of biochar and Se did not result in a significant effect. Huayu 20 indicated different trends. The addition of Se resulted in the transfer of Cd from the roots to the shoots, an increase in the transport coefficient of Cd from roots to leaves, an increase in the accumulation of Cd in peanut leaves, and a significant decrease in the transport coefficient of Cd from leaves to grains. This indicates that Se decreased the movement of Cd to grains. Moreover, Se significantly increased the Se content in peanut roots, stems, leaves, and grains. Foliar Se significantly increased the Se content in the ground and grains of peanuts, while the soil Se majorly accumulated in the roots. This indicated that Se was more easily absorbed through the phloem when applied to the leaves than through the xylem when applied to the soil [49].

To study the mechanism of Se in leaves, we determined the subcellular distribution of Cd in peanut leaves. The addition of Se decreased the percentage of Cd in the cell walls of peanuts and increased the proportion of Cd in the soluble cytoplasm [50]. Spraying Se on leaves had a significantly higher effect than application of Se in the soil did, which was related to the higher accumulation of Se in the leaves during foliar Se spraying. Selenium can alleviate the damage caused by Cd in plants in several ways [51]. In plant cells, Cd binds sulphur-containing ligands, such as GSH and PC [52,53]. The Cd–PC complex is transferred to the vacuole to decrease the damage of Cd to organelles [54]. The synthesis of PC is an important mechanism for plants to detoxify and tolerate Cd. The synthesis and accumulation of PC is strongly dependent on GSH [55]. Under the conditions of Cd-contaminated soil, foliar Se significantly increased the GSH content in peanut leaves, which in turn increased the synthesis of PC in the leaves. Selenium promoted the fixation of Cd in peanut leaves by increasing the content of GSH and PC, thereby decreasing the movement of Cd into the grains.

Further, we performed SEM and EDS analyses of peanut leaves. In plants, Cd-induced reactive oxygen species can directly or indirectly destroy molecular functions. Excessive reactive oxygen species can cause DNA, gene, and protein dysfunction, rupture of the plasma membrane of the cell, and cell death in severe cases [7,56]. Our study showed that the peanut leaves were not under severe Cd stress and thus, cell death was not observed. However, its toxicity can damage organelles and ultrastructures and affect plant physiological activities [57]. Moreover, a comparative analysis of the relative mass and atomic weight of Cd and Se revealed an increase in the relative amount of Se and a decrease in the relative amount of Cd after the addition of Se. This was consistent with previous test results. In addition, Cd affects the absorption of trace elements, such as Mg, Si, S, Mn, and Fe [58,59]. After the addition of Se, the relative quality of these elements in the mesophyll cells was considerably improved, indicating that Se can alleviate the toxicity of Cd by promoting the absorption of trace elements.

Overall, Se plays an important role in alleviating Cd stress in peanuts, which varies depending on the mode of application.

Soil application of Se: (1) soil selenium forms a Cd—Se complex with Cd by affecting the rhizosphere microenvironment and the formation of selenium metabolites, reducing the bioavailability of Cd in the soil; (2) soil Se promotes the absorption of trace elements, such as Mg, Si, S, Mn, and Fe, by plants; (3) the enrichment of Se in peanuts was concentrated in the roots, while the accumulation of Se was in leaves and grains; the detoxification effect of Se in plants and the enrichment of Se in the grains were low.

Foliar application of Se: (1) foliar Se was more conducive to the enrichment of Se in peanut plants and grains, which can effectively promote the growth of peanuts and improve the tolerance of peanuts to Cd; (2) selenium promotes the synthesis of GSH and PC in peanut cells to facilitate Cd to form Cd–PC complexes in the cells, fix them in vacuoles, and decrease the movement of Cd into the grains; (3) foliar Se cannot interact with Cd in soil.

However, the adsorption and complexation of heavy metals by biochar decreased the bioavailability of Cd in the soil, effectively compensating for the shortcomings of foliar Se. Therefore, the combination of biochar and foliar spraying of Se fertilizer was the best treatment to enrich Se and suppress Cd in peanuts.

## 5. Conclusions

In this study, both biochar and Se effectively decreased the absorption of Cd by peanuts and decreased the content of Cd in peanut grains. Moreover, the combined application of biochar and Se further alleviated the stress of Cd on peanuts. The application of Se resulted in the transfer of Cd from the roots to the shoots, an increase in the transport coefficient of Cd from roots to leaves, an increase in the accumulation of Cd in peanut leaves, and a significant decrease in the transport coefficient of Cd from leaves to grains. The addition of Se decreased the percentage of Cd in the cell walls of peanuts and increased the proportion of Cd in the soluble cytoplasm. Under the conditions of Cd-contaminated soil, foliar Se significantly increased the GSH content in peanut leaves, which in turn increased the synthesis of PC in the leaves. Selenium promoted the fixation of Cd in peanut leaves by increasing the content of GSH and PC, thereby decreasing the movement of Cd into the grains. Considering factors such as peanut growth, grain yield, and Cd content in grains, foliar spraying of Se fertilizer combined with biochar was the best treatment for the alleviation of Cd stress among the five treatments. Biochar can adsorb Cd in soil and decrease its bioavailability. Selenium can increase the absorption of trace elements and promote the synthesis of GSH and PC, thereby enhancing the growth of peanuts, reducing the toxicity of Cd, and reducing the movement of Cd into the grains. However, extensive studies are required to understand the other mechanisms of Cd and Se in the soil–plant system before the combined application of biochar and Se can be implemented in contaminated farmlands.

## Figures and Tables

**Figure 1 ijerph-19-03542-f001:**
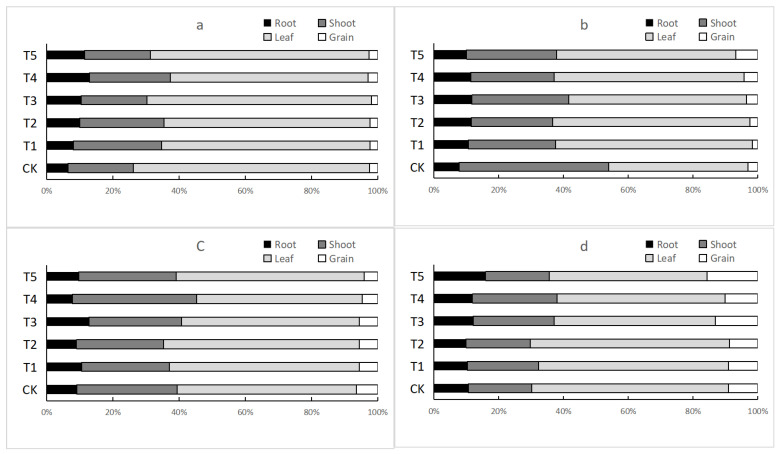
Cadmium accumulation in the roots, stems, leaves, and grains of Huayu 23 and Huayu 20. (**a**) Huayu 23 with high cadmium pollution; (**b**) Huayu 20 with high cadmium pollution; (**c**) Huayu 23 with low cadmium pollution; (**d**) Huayu 20 with low cadmium pollution.

**Figure 2 ijerph-19-03542-f002:**
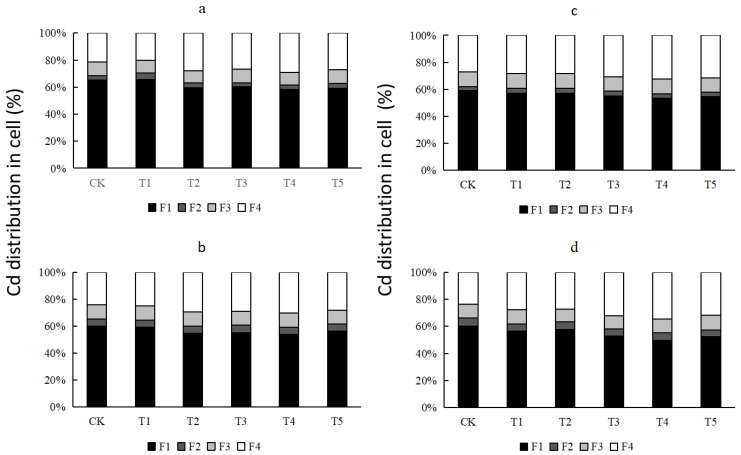
Subcellular distribution of cadmium in the leaves of Huayu 23 and Huayu 20. (**a**) Huayu 23 with high cadmium pollution; (**b**) Huayu 23 with low cadmium pollution; (**c**) Huayu 20 with high cadmium pollution; (**d**) Huayu 20 with low cadmium pollution. F1, F2, F3 and F4 are components of the cell wall, cell nucleus and chloroplast, mitochondria, and soluble cytoplasm, respectively.

**Figure 3 ijerph-19-03542-f003:**
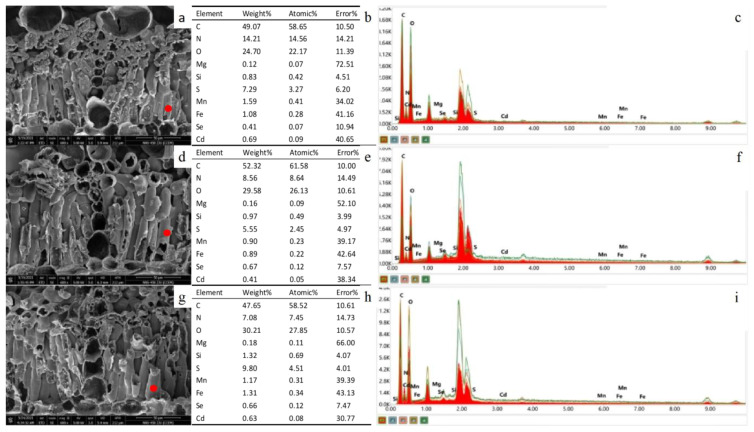
Scanning electron microscope images and energy spectrum analysis of leaves of Huayu 23 under blank (**a**–**c**), single leaf selenium (**d**–**f**), and single soil selenium treatments (**g**–**i**). The red dots in the figure represent the selection points for energy spectrum analysis.

**Table 1 ijerph-19-03542-t001:** Biomass (g) and grain yield (g) of roots and shoots of Huayu 23 and Huayu 20 under different treatments.

Variety	Biochar	Se	Treatments	High Cd			Low Cd		
				Root	Above Ground	Grain	Root	Above Ground	Grain
Huayu 23	no biochar	no	CK	0.54 e	7.39 f	0.43 f	0.80 b	8.11 f	1.07 e
	biochar	no	T1	0.67 d	9.00 e	0.48 e	1.15 b	10.11e	1.21 d
	no biochar	soil	T2	0.83 c	9.33 d	0.76 b	1.13 b	9.34 d	1.37 c
	biochar	soil	T3	1.14 b	11.98 b	0.90 a	1.64 a	10.56 c	1.47 b
	no biochar	foliar	T4	1.32 a	9.77 c	0.51 d	1.10 b	13.49 b	1.37 c
	biochar	foliar	T5	1.33 a	12.28 a	0.63 c	1.92 a	16.79 a	1.71 a
Huayu 20	no biochar	no	CK	0.42 d	6.02 e	0.41 f	0.57 f	4.08 f	1.12 e
	biochar	no	T1	1.15 c	8.69 d	0.50 e	0.62 e	4.80 e	1.31 d
	no biochar	soil	T2	1.25 b	9.44 c	0.72 d	0.87 d	5.57 d	1.42 c
	biochar	soil	T3	1.27 b	10.37 a	1.07 c	1.27 b	7.06 b	2.49 b
	no biochar	foliar	T4	1.24 b	10.23 b	1.34 b	0.91 c	5.87 c	1.42 c
	biochar	foliar	T5	1.43 a	10.40 a	1.79 a	2.05 a	7.58 a	3.78 a
Analysis of variance		Variety (V)		0.000	0.000	0.000	0.001	0.000	0.000
		Biochar (B)		0.000	0.000	0.000	0.000	0.000	0.000
		Se (S)		0.000	0.000	0.000	0.000	0.000	0.000
		V × B		0.000	0.000	0.000	0.790	0.000	0.000
		V × S		0.000	0.214	0.000	0.069	0.000	0.000
		B × S		0.000	0.000	0.000	0.000	0.000	0.000
		V × B × S		0.000	0.000	0.000	0.134	0.000	0.000

Data are presented as the mean (*n* = 3). According to the least significant difference multiple comparison test, different letters in the same column indicate significant differences between treatments of the same peanut variety (*p* ≤ 0.05).

**Table 2 ijerph-19-03542-t002:** Cadmium content (mg·kg^−1^) in the roots, stems, leaves, and grains of Huayu 23 and Huayu 20 under different treatments.

Variety	Biochar	Se	Treatment	High Cd				Low Cd			
				Root	Shoot	Leaf	Grain	Root	Shoot	Leaf	Grain
Huayu 23	no biochar	no	CK	28.12 a	17.48 a	35.25 a	13.83 a	15.42 a	9.27 a	20.06 a	8.19 a
	biochar	no	T1	27.80 a	14.68 b	29.80 b	11.43 b	14.54 a	9.01 ab	19.68 a	7.34 ab
	no biochar	soil	T2	27.24 ab	13.34 c	27.73 c	9.96 c	13.14 b	8.39 ab	19.58 a	6.68 b
	biochar	soil	T3	26.33 b	12.48 c	26.20 d	8.83 d	12.48 b	8.00 b	17.14 b	6.01 bc
	no biochar	foliar	T4	23.02 c	12.57 c	27.42 c	9.20 d	12.68 b	8.33 b	16.48 b	6.16 bc
	biochar	foliar	T5	21.85 c	8.66 d	25.56 e	7.21 e	11.02 c	7.52 b	15.03 c	5.27 c
Huayu 20	no biochar	no	CK	36.18 a	23.28 a	38.40 a	14.76 a	23.35 a	14.15 a	32.62 a	10.20 a
	biochar	no	T1	24.58 b	17.06 b	36.46 b	9.07 b	19.17 b	11.36 b	26.36 b	7.95 b
	no biochar	soil	T2	23.48 c	13.35 c	33.41 c	8.58 c	14.00 c	10.54 c	22.93 c	7.47 bc
	biochar	soil	T3	21.88 d	12.64 c	27.34 e	7.75 de	11.42 e	9.47 d	15.42 e	6.32 d
	no biochar	foliar	T4	23.02 c	12.91 c	28.56 d	7.93 d	13.39 d	9.69 d	17.23 d	7.34 c
	biochar	foliar	T5	13.71 e	10.52 d	20.48 f	7.37 e	10.42 f	7.86 e	15.33 e	5.53 e
Analysis of variance		Variety (V)		0.000	0.000	0.000	0.000	0.000	0.000	0.000	0.000
		Biochar (B)		0.000	0.000	0.000	0.000	0.001	0.000	0.000	0.000
		Selenium (S)		0.000	0.000	0.000	0.000	0.000	0.000	0.000	0.000
		V × B		0.000	0.000	0.000	0.497	0.000	0.000	0.000	0.000
		V × S		0.000	0.001	0.000	0.000	0.000	0.000	0.000	0.000
		B × S		0.000	0.000	0.000	0.000	0.000	0.004	0.000	0.000
		V × B × S		0.000	0.000	0.000	0.000	0.000	0.734	0.000	0.354

Data are presented as the mean (*n* = 3). According to the least significant difference multiple comparison test, different letters in the same column indicate significant differences between treatments of the same peanut variety (*p* ≤ 0.05).

**Table 3 ijerph-19-03542-t003:** Selenium content (mg·kg^−1^) in the roots, stems, leaves, and grains of Huayu 23 and Huayu 20 under different treatments.

Variety	Biochar	Se	Treatments	High Cd				Low Cd			
				Root	Shoot	Leaf	Grain	Root	Shoot	Leaf	Grain
Huayu 23	no biochar	no	CK	0.27 d	0.03 b	0.10 c	0.05 d	0.27 d	0.04 d	0.26 c	0.09 d
	biochar	no	T1	0.27 d	0.03 b	0.10 c	0.05 d	0.29 d	0.04 d	0.21 c	0.12 d
	no biochar	soil	T2	3.63 a	0.13 b	0.20 c	0.44 c	2.52 b	0.13 c	0.39 c	0.59 c
	biochar	soil	T3	2.73 b	0.15 b	0.23 c	0.32 c	2.99 a	0.16 c	0.34 c	0.59 c
	no biochar	foliar	T4	1.41 c	1.36 a	6.60 a	4.02 a	1.29 c	1.88 a	7.07 a	3.33 a
	biochar	foliar	T5	1.47 c	1.17 a	6.02 b	3.41 b	1.29 c	1.11 b	5.02 b	3.21 b
Huayu 20	no biochar	no	CK	0.22 c	0.08 d	0.02 b	0.07 c	0.32 c	0.03 c	0.02 b	0.04 d
	biochar	no	T1	0.12 c	0.07 d	0.02 b	0.03 c	0.03 c	0.04 c	0.02 b	0.02 d
	no biochar	soil	T2	1.66 a	0.96 b	0.15 b	0.34 c	0.79 b	0.29 b	0.49 b	0.85 c
	biochar	soil	T3	1.32 b	0.73 c	0.26 b	0.34 c	1.50 a	0.37 b	0.53 b	0.88 c
	no biochar	foliar	T4	1.33 b	1.71 a	6.41 a	3.11 a	1.81 a	2.38 a	5.26 a	4.66 a
	biochar	foliar	T5	1.39 b	1.55 a	5.94 a	2.28 b	1.74 a	2.43 a	4.98 a	4.03 b
Analysis of variance		Variety (V)		0.000	0.000	0.581	0.000	0.000	0.000	0.004	0.000
		Biochar (B)		0.000	0.416	0.279	0.000	0.021	0.005	0.001	0.000
		Selenium (S)		0.000	0.000	0.000	0.000	0.000	0.000	0.000	0.000
		V × B		0.092	0.018	0.836	0.671	0.711	0.000	0.006	0.001
		V × S		0.000	0.000	0.937	0.000	0.000	0.000	0.001	0.000
		B × S		0.000	0.451	0.169	0.000	0.000	0.000	0.000	0.000
		V × B × S		0.012	0.214	0.987	0.423	0.152	0.000	0.003	0.000

Data are presented as the mean (*n* = 3). According to the least significant difference multiple comparison test, different letters in the same column indicate significant differences between treatments of the same peanut variety (*p* ≤ 0.05).

**Table 4 ijerph-19-03542-t004:** Cadmium transport coefficients of Huayu 23 and Huayu 20 from roots to stems, stems to leaves, and leaves to grains under different treatments.

Treatments	Huayu 23					Huayu 20				
	TFRoot—Shoot	TFShoot—Leaf	TFRoot—Grain	TFShoot—Grain	TFLeaf—Grain	TFRoot—Shoot	TFShoot—Leaf	TFRoot—Grain	TFShoot—Grain	TFLeaf—Grain
High Cd										
CK	0.62 a	2.02 b	0.49 a	0.79 b	0.39 a	0.64 c	1.65 d	0.41 b	0.63 b	0.38 a
T1	0.53 bc	2.03 b	0.34 c	0.69 c	0.38 a	0.69 b	2.14 b	0.35 c	0.61 b	0.25 d
T2	0.49 cd	2.09 b	0.32 c	0.70 c	0.36 b	0.57 d	2.50 a	0.37 c	0.70 a	0.26 d
T3	0.47 d	2.10 b	0.27 d	0.58 d	0.34 c	0.58 d	2.16 b	0.37 c	0.64 b	0.28 c
T4	0.55 b	2.18 b	0.50 a	0.78 b	0.34 c	0.56 d	2.22 b	0.34 c	0.47 c	0.28 c
T5	0.40 e	2.96 a	0.46 b	1.15 a	0.28 d	0.77 a	1.95 c	0.54 a	0.70 a	0.36 b
Low Cd										
CK	0.60 b	2.16 ab	0.53 c	0.88 a	0.41 a	0.61 c	2.31 a	0.44 b	0.72 b	0.31 c
T1	0.62 ab	2.19 ab	0.42 b	0.69 b	0.37 ab	0.59 c	2.32 a	0.55 a	0.65 b	0.30 c
T2	0.64 ab	2.35 a	0.58 ab	0.92 a	0.34 b	0.75 b	2.18 a	0.57 a	0.82 a	0.33 c
T3	0.64 ab	2.15 ab	0.60 a	0.80 ab	0.35 ab	0.83 a	1.63 c	0.53 a	0.71 b	0.41 a
T4	0.66 ab	1.98 b	0.42 c	0.70 b	0.37 ab	0.72 b	1.78 bc	0.39 c	0.71 b	0.43 a
T5	0.68 a	2.00 b	0.46 c	0.72 b	0.35 ab	0.75 b	1.96 b	0.55 a	0.67 b	0.36 b

Data are presented as the mean (*n* = 3). According to the least significant difference multiple comparison test, different letters in the same column indicate significant differences between treatments of the same peanut variety (*p* ≤ 0.05).

**Table 5 ijerph-19-03542-t005:** Glutathione content (µmol·g^−1^) in Huayu 23 and Huayu 20 under different treatments.

Treatments	Huayu 23 High Cd	Huayu 23 Low Cd	Huayu 20 High Cd	Huayu 20 Low Cd
CK	0.64 b	0.56 c	0.84 b	0.56 c
T1	0.66 b	0.53 c	0.87 b	0.57 c
T2	0.73 b	0.67 b	0.82 b	0.74 b
T3	0.72 b	0.63 b	0.80 b	0.68 bc
T4	0.95 a	0.74 a	1.12 a	0.95 a
T5	0.74 b	0.73 a	1.07 a	0.75 b

Data are presented as the mean (*n* = 3). According to the least significant difference multiple comparison test, different letters in the same column indicate significant differences between treatments of the same peanut variety (*p* ≤ 0.05).

**Table 6 ijerph-19-03542-t006:** Phytochelatin content (µmol·g^−1^) of Huayu 23 and Huayu 20 under different treatments.

Treatments	Huayu 23 High Cd	Huayu 23 Low Cd	Huayu 20 High Cd	Huayu 20 Low Cd
CK	16.42 d	15.60 e	24.48 e	16.49 e
T1	15.30 e	15.35 f	24.55 e	17.71 d
T2	29.51 c	19.44 d	29.43 c	22.63 c
T3	30.63 b	25.36 c	27.58 d	22.73 c
T4	38.47 a	29.51 a	39.88 a	28.55 a
T5	30.63 b	28.32 b	39.35 b	25.64 b

Data are presented as the mean (*n* = 3). According to the least significant difference multiple comparison test, different letters in the same column indicate significant differences between treatments of the same peanut variety (*p* ≤ 0.05).

**Table 7 ijerph-19-03542-t007:** Correlation analyses of Se and Cd content in Huayu 23 and Huayu 20 roots, stems, leaves, and grains under different treatments.

Treatments	Root	Shoot	Leaf	Grain
Huayu 23				
High Cd	−0.057	−0.729 **	−0.476 *	−0.636 **
Low Cd	−0.534 *	−0.401	−0.745 **	−0.658 **
Huayu 20				
High Cd	−0.636 *	−0.784 **	−0.715 **	−0.472 *
Low Cd	−0.829 **	−0.678 **	−0.645 **	−0.558 *

* *p* ≤ 0.05 and ** *p* ≤ 0.01 indicate statistical significance.

## Data Availability

All data generated or analyzed during this study are included in this published article.

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
