# Peer review of "Foliar Spraying of Selenium Combined with Biochar Alleviates Cadmium Toxicity in Peanuts and Enriches Selenium in Peanut Grains"

_ijerph, 2022, doi:10.3390/ijerph19063542_

Round 1

Reviewer 1 Report

My suggestions can be found in the attached file.

Reviewer 2 Report

Review of manuscript: “Foliar spraying of selenium combined with biochar alleviates cadmium toxicity in peanuts and enriches selenium in peanut grains” by Shiwei Shao et al.

The work is presented well, with clear writing and organization. It also addresses an interesting and important issue. I’ll list a few minor issues to address.

Line 8 – I would reword the first sentence to: “Cadmium (Cd) pollution in soil, particularly in peanut production, is a problem that has attracted global concern and needs solutions urgently.”

Line 11 – The scientific/Latin name for peanut should be presented in italics and cultivar names are conventionally presented with single quotes (e.g. ‘Huayu 23’)

Line 42 – See previous point.

Line 43 – Can the authors provide a reference for this production statistic?

Line 110 – The author write that the experiment was conducted in “the scientific research intelligent greenhouse.” If these descriptors for the greenhouse are important, then more description of what is meant should be provided. Or, the authors could provide a reference where more information can be found about the facility.

Line 124 – Add a space after “0.5.”

Line 322 – Can the authors add a label to the y axes of these four graphs? And is the y axis in graph “b” supposed to be 50% as the maximum? I’m a bit confused by this presentation.

Line 366 – The clarity of the table component of this figure could be improved, possibly by eliminating the alternate shading of rows. Also, there are missing spaces before the parentheses in the figure caption.

Line 449 – Does this single sentence need to be a paragraph or can it be combined with the paragraph below?

Reviewer 3 Report

I like the study, I find it valuable and new in case of  subcellular structures and SEM and EDS. I find the data collected in the manuscript as interesting. I have 15 comments, this is a lot however most of them a debatable. The rest are the results of some weakness of the manuscript. I advise to accept the manuscript after corrections.     

I have following comments:

  1. Please change manuscript title so it match to the text of the manuscript. Foliar spraying of selenium combined with biochar is one of fifth combination (six treatments) in yours study. So why do you use only them in the title? Also the second part of the tittle i.e. “and enriches selenium in peanut grains” needs to be deleted. Increase of Se content is very positive results of yours cadmium detoxification measures. However if you wanted to increase the Se content in the peanut plants contamination of soil with cadmium was not needed.
  2. The aim. Lines 85-92.

2.1. There is nothing about testing the effectiveness of detoxification measures.

2.2. “Identifying the best combination of biochar and Se fertiliser” I understand this sentence as testing different doses of biochar and Se. This is not carried on in the study.

2.3.  There is nothing in the aim about studying the Cd content in plant parts i.e. root, above ground parts and grains.

2.4. There is nothing about biomass.

2.5.  Lines 89-90. The aim for study subcellular is not clear for me.

2.6. Redefine aim for selenium enrichment in peanut grains.   

  1. Why did you take farmland (arable??) soil Chengyang District, Qingdao? Why didn’t you take a standard substrate from gardener shop? What was the crop rotation in the arable land before the soil was taken? What was the selenium content in this soil? What was the cadmium content in the soil?
  2. Why did you soak the seeds in H2O2 solution? Why did you germinate in an artificial conditions? What was the source of seeds? Is peanut cultivated in Chengyang District?
  3. Did you see any physiological diseases ? Or any growth disorders due to so high cadmium contamination?
  4. Why did you take peanut plant? Why did you take two varieties? How many of them did you take into account? What are differences between them? Only lower and higher bioaccumulation?
  5. Why didn’t you take into account different species instead of different varieties?
  6. Please explain why did you chose cadmium contamination levels at 5 and 20 mg*kg -1 . Are such concertation of cadmium noticed elsewhere? Consider use lower and higher Cd, or 5 and 20 mg*kg -1 because 5 mg*kg -1 of cadmium is also very high. Colling this concentration as low is misleading.
  7. Table 4. Please add TFroot-grain and TFroot-leafe, This data would be interesting for health specialists.
  8. Table 7. Delate the subtable comment i.e. “Data are presented as the mean + standard error (n=3). By the way why do you write n=3 if you have for replicates? Line 122 “Each treatment was repeated four times per group.
  9. Why did you use perichloric acid for plant material mineralisation? What did you wanted to reach? Instead overnight (for me not clear) write how many hours line 140.
  10. Why did you chose leaves to study subcellular cadmium content? Please explain. Are leaves or shoots fodder or food to any life?
  11. What did you wanted to reach using SEM and EDS? What is relative atomic weight of Cd to Se.
  12. Please consider if the control (I called it the zero control 0CK) shouldn’t be the soil of Qingdao not contaminated (I guess?). This is in contents of relative atomic weight, the biomass, the translocation factor, the increase of selenium content and finally the cadmium content?
  13. Please think over the order, the structure of the manuscript. About the tittle and aim I have already written. The results and the discussion. Using statistical analysis is a prat of discussion, so discussion is in Discussion Chapter and in Results Chapter. Lines of 349 to 356 now belongs to Results Chapter in my opinion should belong to the Materials and Methods Chapter. The Conclusions Chapter is short. “Cd toxicity to human health” line 477 was not studied so delate “to human health”. Also “Se deficiency in humans” was not studied so delete it. I don’t find important lines 479 to The lines may be deleted. There is no singular sentence of cadmium transport and distribution in plant parts. The same is with the cadmium content in subcellular components. It is new and interesting part of the study please add this conclusions to the Chapter.

Reviewer 4 Report

Autors investigated the effects Se application methods combined with biochar on the accumulation of Cd and Se on two varieties of peanut (Arachis hypogaea Linn.) Huayu 23 and Huayu 20. In addition, the role of Se in alleviating Cd toxicity in peanuts was studed. The study indicated that both Se and biochar decreased the Cd content in peanuts and alleviated Cd toxicity. The study proved as well that biochar can decrease the absorption of Cd from the soil, while Se can increase the accumulation of Cd in cell vacuoles by increasing glutathione and phytochelatin to decrease the movement of Cd into the grains. The results indicated that the combined application of foliar Se and biochar can effectively promote the enrichment of Se in peanuts and suppress Cd toxicity. The results of these studies are very interesting, promising and  useful. The methods presented are clear, repeatable and possible to implement. Nevertheless, the manuscript needs to be corrected. Changes and suggestions would be found in the attached PDF-file, in the specific comments for all sections of the paper.

Round 2

Reviewer 1 Report

look into the attached file.

Author Response

Thank you for the suggestion.Tables 1, 2, 3 have been modified as required (Line 201, page 5; line 285, page 7; line 290, page 7).